# Benign, Tempered, or Catastrophic: A Taxonomy of Overfitting

**Neil Mallinar**[*]
UC San Diego
nmallina@ucsd.edu

**James B. Simon**[*]
UC Berkeley
james.simon@berkeley.edu

**Amirhesam Abedsoltan**
UC San Diego
aabedsoltan@ucsd.edu

**Parthe Pandit**
UC San Diego
parthepandit@ucsd.edu

**Mikhail Belkin**
UC San Diego
mbelkin@ucsd.edu

**Preetum Nakkiran**
Apple & UC San Diego
preetum@apple.com

## Abstract

The practical success of overparameterized neural networks has motivated the recent scientific study of *interpolating methods*, which perfectly fit their training data. Certain interpolating methods, including neural networks, can fit noisy training data without catastrophically bad test performance, in defiance of standard intuitions from statistical learning theory. Aiming to explain this, a body of recent work has studied *benign overfitting*, a phenomenon where some interpolating methods approach Bayes optimality, even in the presence of noise. In this work we argue that while benign overfitting has been instructive and fruitful to study, many real interpolating methods like neural networks *do not fit benignly*: modest noise in the training set causes nonzero (but non-infinite) excess risk at test time, implying these models are neither benign nor catastrophic but rather fall in an intermediate regime. We call this intermediate regime *tempered overfitting*, and we initiate its systematic study. We first explore this phenomenon in the context of kernel (ridge) regression (KR) by obtaining conditions on the ridge parameter and kernel eigenspectrum under which KR exhibits each of the three behaviors. We find that kernels with powerlaw spectra, including Laplace kernels and ReLU neural tangent kernels, exhibit tempered overfitting. We then empirically study deep neural networks through the lens of our taxonomy, and find that those trained to interpolation are tempered, while those stopped early are benign. We hope our work leads to a more refined understanding of overfitting in modern learning.

## 1 Introduction

In the last decade, the dramatic success of overparameterized deep neural networks (DNNs) has inspired the field to reexamine the theoretical foundations of generalization. Classical statistical learning theory suggests that an algorithm which *interpolates* (i.e. perfectly fits) its training data will typically *catastrophically overfit* at test time, generalizing no better than a random function.

Figure 1c illustrates the catastrophic overfitting classically expected of an interpolating method. Defying this picture, DNNs can interpolate their training data and generalize well nonetheless [Neyshabur et al., 2015, Zhang et al., 2017], suggesting the need for a new theoretical paradigm within which to understand their overfitting.

---

[*]Co-first authors.

36th Conference on Neural Information Processing Systems (NeurIPS 2022).

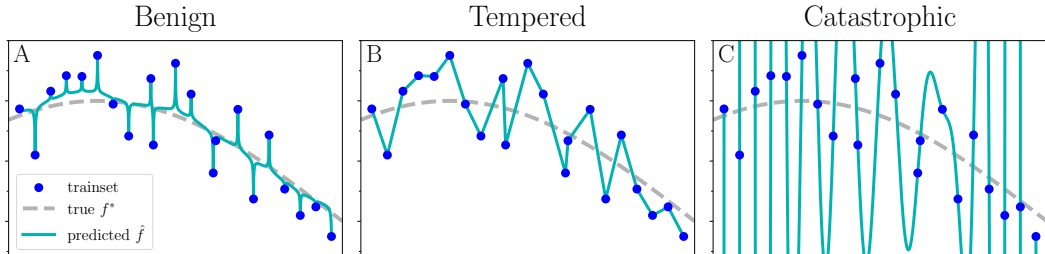

Figure 1: **As** $n \rightarrow \infty$**, interpolating methods can exhibit three types of overfitting.** **(A)** In *benign overfitting*, the predictor asymptotically approaches the ground-truth, Bayes-optimal function. Nadaraya-Watson kernel smoothing with a singular kernel, shown here, is asymptotically benign. **(B)** In *tempered overfitting*, the regime studied in this work, the predictor approaches a constant test risk greater than the Bayes-optimal risk. Piecewise-linear interpolation is asymptotically tempered. **(C)** In *catastrophic overfitting*, the predictor generalizes arbitrarily poorly. Rank-$n$ polynomial interpolation is asymptotically catastrophic.

This need motivated the identification and study of *benign overfitting* using the terminology of [Bartlett et al., 2020] (also called "harmless interpolation" [Muthukumar et al., 2020]), a phenomenon in which certain methods that perfectly fit the training data still approach *Bayes-optimal* generalization in the limit of large trainset size. Intuitively speaking, benignly-overfitting methods fit the target function globally, yet fit the noise only locally, and the addition of more label noise does not asymptotically degrade generalization. Figure 1a illustrates a simple method that is asymptotically benign[2]. The study of benign overfitting has proven fruitful, leading to rich mathematical insights into high-dimensional learning[3], and benign overfitting is certainly closer to the real behavior of DNNs than catastrophic overfitting.

That said, it requires only simple experiments to reveal that many standard DNNs *do not overfit benignly*: when training on noisy data, DNNs do not diverge catastrophically, but *neither* do they approach Bayes-optimal risk. Instead, they converge to a predictor that is neither catastrophic nor optimal but rather somewhere in between, with error that increases as the noise in the data increases. Figure 2 depicts such an experiment: a ResNet is trained on a binary variant of CIFAR-10 with varying amounts of training label noise, and with increasing sample size $n$. We see from Figure 2 that greater train noise indeed results in greater test error, and this test error persists even as $n$ grows, converging to a non-zero asymptotic value[4]. This is unlike "benign overfitting," which would produce an asymptotically-optimal predictor at all non-trivial noise levels (depicted in blue in Figure 2). This suggests that, in the search for a paradigm to understand modern interpolating methods, we should identify and study a regime intermediate between benign and catastrophic.

## 1.1 Summary of Contributions

In this work we formally identify an intermediate regime between benign and catastrophic overfitting. We call this intermediate behavior *tempered overfitting* because the noise's harmful effect is tempered but still nonzero. We find that both DNNs trained to interpolation and (ridgeless) kernel regression (KR) using certain common kernels fall into this intermediate regime *even as the number of training examples $n$ approaches infinity*, as do common methods like 1-nearest-neighbors and piecewise-linear interpolation (as in Figure 1b). Our tempered regime completes the taxonomy of overfitting: essentially any learning procedure is either benign, tempered, or catastrophic in the asymptotic limit.

---

[2] As a first hint that important practical methods may not be benign (at least in low dimension), note that, in order to be benign, the predicted function in Figure 1a *has* to take this spiky shape. On the other hand, very wide and deep neural network may indeed be spiky Radhakrishnan et al. [2022].

[3] A partial list of works here include Advani and Saxe [2017], Bahri et al. [2020, 2021], Bartlett et al. [2020, 2021], Belkin et al. [2018a,b, 2019a], Cao et al. [2022], Chatterji and Long [2021], Chatterji et al. [2021], d'Ascoli et al. [2020], Frei et al. [2022], Goldt et al. [2019], Hastie et al. [2019], Koehler et al. [2021], Liang and Rakhlin [2018, 2020], Mei and Montanari [2019], Muthukumar et al. [2020], Rakhlin and Zhai [2019], Tsigler and Bartlett [2020], Zhang et al. [2017, 2021].

[4] It is well-known and is perhaps unsurprising that interpolating DNNs are harmed by label noise (e.g. Zhang et al. [2017]); our new observation is that this persists *even as $n \rightarrow \infty$*.

We begin in Section 2 with preliminaries, formal definitions of the three regimes, and a taxonomy of some common ML methods according to these regimes. In Section 3, we study these three regimes for kernel regression (KR). Using recent spectral theories characterizing the expected test error of KR, we obtain conditions on the ridge parameter and kernel eigenspectrum under which KR falls into each of the three regimes. Importantly, we find that ridgeless kernels with powerlaw spectra, including the Laplace kernel and ReLU fully-connected neural tangent kernels (NTKs), are asymptotically *tempered*, not benign. We confirm our theory with experiments on synthetic data. In Section 4, we empirically study overfitting for DNNs. We give evidence that standard DNNs trained to interpolation exhibit tempered overfitting, not benign overfitting, motivating the further study of tempered overfitting in the pursuit of understanding modern machine learning methods. We additionally study the time dynamics of overfitting, and the effect of early-stopping. We conclude with discussion in Sections 5 and 6.

## 1.2 Relation to Prior Work

Our work is inspired by recent developments in the theory and empirics of interpolating methods [Bartlett et al., 2020, Belkin et al., 2018a, 2019c,d, Devroye et al., 1998, Ji et al., 2021, Koehler et al., 2021, Liang and Rakhlin, 2018, 2020, Muthukumar et al., 2020, Rakhlin and Zhai, 2019, Tsigler and Bartlett, 2020]. Many of these works prove that certain interpolating estimators are statistically consistent in certain settings, demonstrating benign overfitting. In contrast, we argue that a wide range of interpolators, including DNNs used in practice, are *not benign*. Our work is compatible with prior work because we consider different settings—erring more on the side of realism— which leads us to different conclusions. The empirical observation that interpolating DNNs for classification are inconsistent, and thus not benign, was made in Nakkiran and Bansal [2020], which in part inspired the present work.

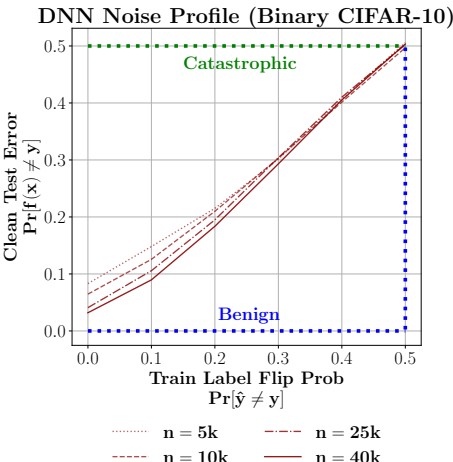

Figure 2: **DNNs trained on image data exhibit *tempered overfitting*, not benign overfitting.** Curves show test classification error vs. training label flip probability for a Wide ResNet $28 \times 10$ trained to interpolation on binary CIFAR-10 (animals vs. vehicles) for different training sizes $n$. These curves are *noise profiles*, as discussed in Section 2.

Briefly, many prior works assume that the target task always lies in "high enough dimension" relative to the number of samples $n$, while our work considers the limit $n \to \infty$ for tasks of fixed dimension $d$, the realistic asymptotic in practice. For example, in the linear regression setting, Bartlett et al. [2020] highlight that benign overfitting occurs most robustly when *input dimension grows faster than the sample size*. Other works explicitly scale the ambient problem dimension and sample size to infinity together at a proportional rate [Hastie et al., 2019, Liang and Rakhlin, 2018, Mei and Montanari, 2019]. This joint scaling is also often considered in statistical physics approaches to learning dynamics (see Zdeborová and Krzakala [2016] and references therein). Taking a different approach, Frei et al. [2022] prove that interpolating two-layer networks can achieve close to optimal test error on certain distributions, but require an assumption that $n \leq \Omega(d)$. And Koehler et al. [2021] state a generalization bound for interpolators that decays to 0 with $n$, but also requires $n \leq d$. In summary, a bulk of prior benign overfitting results apply in the regime where $n$ is large, but still restricted to be smaller than the dimension of the problem, whereas we do not consider such restrictions.

Our work is also compatible with prior works which observe that excess risk of certain interpolating methods decays with ambient dimension, interpreted as high-dimensional problems enjoying a "blessing of dimensionality." For example, the simplicial interpolation scheme of Belkin et al. [2018a] has excess risk that decays as $O(2^{-d})$ for ambient dimension $d$, and Rakhlin and Zhai [2019] show that kernel ridgeless regression with the Laplace kernel is inconsistent in any fixed dimension, but with a lower bound on risk that decays with dimension. We find an explicit result to this effect when studying KR: for Laplace kernels and ReLU NTKs, asymptotic excess mean squared error decays

like $\Theta(1/d)$. These phenomena support our claim that many interpolating methods are *tempered* on real distributions, which have fixed dimension.

## 2 The Three Types of Overfitting

Here we formally present our taxonomy of overfitting, delineating the three types of asymptotic behaviors which learning procedures can exhibit.

### 2.1 Definitions

We consider a fairly generic in-distribution supervised learning setting. For simplicity, we present definitions for regression, but these are readily extended to classification. We wish to learn a function $\widehat{f} : \mathcal{X} \to \mathbb{R}$ from a size-$n$ dataset of i.i.d. samples $\mathcal{D}_n \equiv \{(x_i, y_i)\}_{i=1}^n \sim \mathcal{D}$, where $\mathcal{D}$ is a joint distribution over $\mathcal{X} \times \mathbb{R}$, and $\mathcal{X}$ is the input domain. We shall generally assume nonzero target noise, with $\mathrm{Var}[y_i|x_i] > 0$. We evaluate the generalization performance of $\widehat{f}$ by the *mean squared error* (MSE): $R(\widehat{f}) := \mathbb{E}_{x,y\sim\mathcal{D}}\left[(\widehat{f}(x) - y)^2\right]$. The Bayes-optimal regression function is given by $f^* := \mathrm{argmin}_f R(f)$, where the minimization is over all measurable functions, and has risk $R^*$ which is called the *irreducible risk*. The *excess risk* of any function $\widehat{f}$ is given by $\overline{R}(\widehat{f}) := R(\widehat{f}) - R^*$. We say an estimator achieves *interpolation* if $\widehat{f}_n(x_i) = y_i$ for all $(x_i, y_i) \in \mathcal{D}_n$. These definitions are readily generalized to classification, using classification error in place of MSE.

**Learning Procedure.** The objects of study in our taxonomy are *learning procedures*. Our definition of a learning procedure is quite general, allowing discussion of methods from DNNs to 1-nearest-neighbors. Informally, a learning procedure is simply a specification of which model to output on a given train set of a given size.

Formally, a *learning procedure* $\mathcal{A} := \{A_n\}_n$ is a sequence of (potentially stochastic) functions, indexed by sample size $n \in \mathbb{N}$. At each $n$, the function $A_n : \mathcal{D}_n \mapsto \widehat{f}_n$ inputs a train set $\mathcal{D}_n$ and outputs a "model" $\widehat{f}_n : \mathcal{X} \to \mathbb{R}$. Note that this $n$-dependence allows learning procedures to be *non-uniform*, varying the learning algorithm with sample size $n$. For example, it allows procedures which scale up a DNN or narrow a kernel bandwidth as $n$ grows. The *expected risk* of a learning procedure on $n$ examples from distribution $\mathcal{D}$ is $\mathcal{R}_n := \mathbb{E}_{A_n,\mathcal{D}_n}[R(A_n(\mathcal{D}_n))]$.

### 2.2 The Taxonomy

We shall categorize learning procedures in terms of their *asymptotic expected risk*. We handle regression and $K$-class classification settings separately, due to their different loss scalings. As the number of samples $n \to \infty$, the sequence of expected risks $\{\mathcal{R}_n\}_n$ can behave in three different ways, as listed in Table 1. These three limiting behaviors define our taxonomy.

|  | **Regression** | **Classification** |
|---|---|---|
| **Benign** | $\lim_{n\to\infty} \mathcal{R}_n = R^*$ | $\lim_{n\to\infty} \mathcal{R}_n = R^*$ |
| **Tempered** | $\lim_{n\to\infty} \mathcal{R}_n \in (R^*, \infty)$ | $\lim_{n\to\infty} \mathcal{R}_n \in (R^*, 1 - \frac{1}{K})$ |
| **Catastrophic** | $\lim_{n\to\infty} \mathcal{R}_n = \infty$ | $\lim_{n\to\infty} \mathcal{R}_n = 1 - \frac{1}{K}$ |

Table 1: **Our taxonomy of (over)fitting.**

There is technically a fourth option – that the limit does not exist – but, to our knowledge, this does not describe any non-pathological algorithms. All together, this set of behaviors is exhaustive, describing every possible learning procedure. Note that the definitions for classification and regression are identical, except for the bounds at $\infty$ replaced by the error of the predictor choosing a uniformly[5] random label $(1 - \frac{1}{K})$.

---

[5]We assume balanced classes throughout, for notational simplicity. The definitions can be modified appropriately for imbalanced classes.

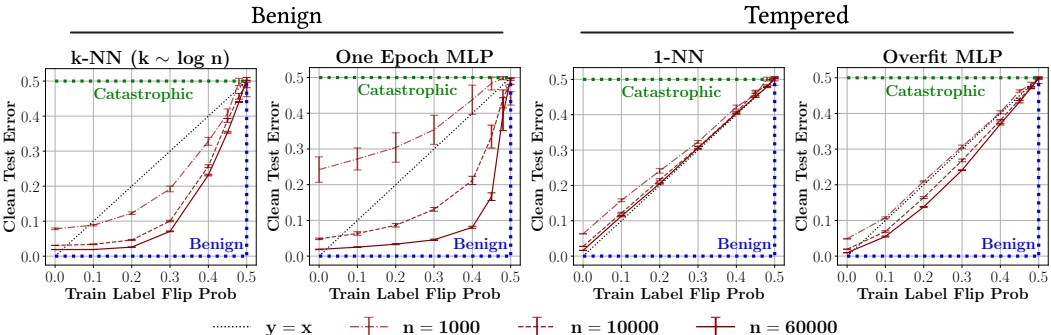

Figure 3: **Examples of Benign and Tempered Fitting.** Noise profiles for several different methods on the Binary-MNIST classification task, showing clean test error as a function of train label noise as the train set $n$ grows. Left: two methods which exhibit benign overfitting, with performance converging to Bayes optimal as $n \to \infty$. Right: two methods which exhibit tempered overfitting, with test error that remains bounded away from 0. The two "tempered" methods here are interpolating, while the two "benign" methods are not. Both the benign and the tempered MLP use identical architectures; one is trained for one epoch, and the other trained to interpolation. Details in Appendix C.3.

| Benign | Tempered | Catastrophic |
|---|---|---|
| • Early-stopped DNNs | • **Interpolating DNNs** | • **Gaussian KR** |
| • KR with ridge | • **Laplace KR** | • Critically-parameterized regression |
| • $k$-NN ($k \sim \log n$) | • **ReLU NTKs** | |
| • Nadaraya-Watson kernel smoothing with Hilbert kernel | • $k$-NN (constant $k$) | |
| | • Simplicial interpolation | |

Table 2: A taxonomy of models under the three types of fitting identified in this work. **BOLD** are results from our work, others are known or folklore results.

## 2.3 Noise Profiles

To study asymptotic risk, we use a tool we call a *noise profile*. A noise profile characterizes the *sensitivity* of a learning procedure to noise in the training set. For a given learning procedure $\mathcal{A}$ and data distribution $\mathcal{D}$, a *noise profile* $\mathsf{P}_{\mathcal{A}}$ describes how the asymptotic risk varies with respect to $\sigma$, the level of artificial noise added to training targets. Formally, the noise profile $\mathsf{P}_{\mathcal{A}}$ is $\mathsf{P}_{\mathcal{A}}(\sigma) = \lim_{n \to \infty} \mathbb{E}_{A_n; \mathcal{D}_n(\sigma)} R(A_n(\mathcal{D}_n(\sigma)))$, where $\mathcal{D}_n(\sigma)$ denotes $n$ i.i.d. samples from the distribution $\mathcal{D}$, with $\sigma$-level of label noise added. The label noise $\sigma \in \mathbb{R}$ denotes a kind of noise that depends on the problem setting: $\sigma$ is the variance of additive Gaussian noise for regression settings, or the label-flip probability for classification settings. We cannot empirically evaluate the $n \to \infty$ limit exactly, so we estimate it using asymptotics from finite but large sample sizes. As shown in Figure 2, noise profiles are easily plotted and reveal at a glance whether an learning procedure is benign, tempered, or catastrophic.

## 2.4 Applying Our Taxonomy

Our taxonomy handles general learning procedures, even those which do not interpolate their train sets[6]. Thus, we can apply it to describe many existing methods in machine learning. In doing so, we *refine the language* around statistical consistency: many methods were known to be statistically inconsistent (i.e. not benign), but we highlight that there are two distinct ways to be inconsistent: tempered and catastrophic.

To illustrate our taxonomy, we give several examples of known results in Table 2. Any statistically consistent method is by definition *benign*: this includes non-interpolating methods such as early-

---

[6]We use "overfitting" to describe interpolating methods, and "fitting" to describe general methods.

stopped DNNs [Ji et al., 2021] and $k$-nearest-neighbors with $k \sim \log n$ [Chaudhuri and Dasgupta, 2014, Cover and Hart, 1967], as well as interpolating methods such as Nadarya-Watson kernel smoothing [Devroye et al., 1998] and k-NN schemes [Belkin et al., 2018a, 2019c].

Many catastrophic learning procedures are also known. Parametric models which are "critically-parameterized" (i.e. at the double descent peak) overfit catastrophically: this includes random feature regression with number of features $p = n$ [Mei and Montanari, 2019], and, more generally, for a broad class of linear and random feature methods [Holzmüller, 2021]. Generic empirical-risk-minimization for a hypothesis class with VC-dimension $> n$ will also overfit catastrophically in the worst case. It turns out that RBF kernel regression under natural assumptions also catastrophically overfits, as we show in Section 3.

Finally, many methods, from classical to modern, exhibit tempered overfitting. First, it is well-known that 1-nearest-neighbors (1-NN) converges to an asymptotic risk that is finite but bounded away from Bayes optimal[7], which is tempered behavior. Further, any "underfitting" method, such as empirical risk minimization over a hypothesis class with VC-dimension $\ll n$, will be tempered if the hypothesis class does not contain the ground-truth function. In Section 4, we empirically demonstrate that many standard DNNs exhibit tempered overfitting when trained to interpolation. We complement this with theoretical results in Section 3, showing that kernel regression with the Laplace kernel, as well as with the ReLU NTK (which describes the training of an infinite-width ReLU fully-connected network [Jacot et al., 2018]), exhibits tempered overfitting. Our new category of tempered overfitting is thus not merely a theoretical possibility but in fact captures many natural and widely-used learning methods.

In Figure 3 we demonstrate our taxonomy experimentally for two benign methods ($k$-NN and early-stopped MLPs) and two tempered methods (1-NN and interpolating MLPs) on a binary classification version of MNIST, with varying noise in the train labels. We plot test classification error on the clean test set against the proportion of flipped labels in the training set. As $n$ grows, the benign methods approach zero test error even at nonzero train noise, while the tempered methods converge to a test error bounded away from zero.

## 3  Overfitting in Kernel Regression

We begin with a study of *kernel regression* (KR), a widely-used nonparameteric learning algorithm which, we will see, is sufficiently rich to exhibit all three regimes of overfitting, yet sufficiently simple that this can be shown analytically. Theoretical interest in this algorithm has increased significantly in recent years due to the discovery that trained DNNs converge to ridgeless KR in the infinite-width and infinite-time limit [Jacot et al., 2018], implying that insights into KR simultaneously shed light on overparameterized DNNs. The overparameterized linear regression setting of Bartlett et al. [2020] is equivalent to KR, and we will make a direct comparison with their results at the end of the section.

KR is fully specified by a positive-semidefinite kernel function $K : \mathbb{R}^d \times \mathbb{R}^d \to \mathbb{R}$ and a ridge parameter $\delta \geq 0$. We allow the training set $\mathcal{D}_n$ to contain $n$ samples $(x_i, y_i) \sim \mathcal{D}$, and we assume that $y_i = f^*(x_i) + \eta_i$ with true function $f^*$ and noise $\eta_i \sim \mathcal{N}(0, \sigma^2)$. KR returns the predicted function $\widehat{f}$ given by

$$\widehat{f}(x) = K(x, \mathcal{D}_n) \left( K(\mathcal{D}_n, \mathcal{D}_n) + \delta \mathbf{I}_n \right)^{-1} \mathcal{Y}, \tag{1}$$

where $K(\mathcal{D}_n, \mathcal{D}_n)$ is the "data-data kernel matrix" with components $K(\mathcal{D}_n, \mathcal{D}_n)_{ij} = K(x_i, x_j)$, $K(x, \mathcal{D})$ is a row vector with components $K(x, \mathcal{D})_i = K(x, x_i)$, and $\mathcal{Y}$ is a column vector of targets.

Existing literature provides examples of KR exhibiting all three asymptotic behaviors. Benign overfitting has been analyzed for overparameterized linear regression [Advani and Saxe, 2017, Bartlett et al., 2020, Belkin et al., 2019b, Hastie et al., 2019, Muthukumar et al., 2020], a special case of KR. It is well-known that KR with a positive ridge value (with appropriate scaling conventions) is consistent and thus benign [Christmann and Steinwart, 2007]. Furthermore, in 1D, a Laplace kernel approaches piecewise linear interpolation as $n \to \infty$ [Belkin et al., 2018a], which is easily shown to exhibit tempered overfitting, and Rakhlin and Zhai [2019] proved that the Laplace kernel does not overfit benignly in (fixed) dimension greater than one (though did not say whether it was in fact tempered or catastrophic). Finally, it is known in experimental folklore (though not theoretically, to our knowledge) that KR with a Gaussian kernel and zero ridge tends to yield poorly-conditioned

---

[7]In fact, the excess test MSE converges to the variance of the observation noise.

kernel matrices and catastrophic behavior. Here we derive fairly general conditions under which KR falls into each regime, solidifying these various observations into a unified picture.

As our chief tool for obtaining these conditions, we use a recently-derived closed-form approximation for the expected test MSE of KR [Bordelon et al., 2020, Canatar et al., 2021, Jacot et al., 2020, Simon et al., 2021]. By simply taking the $n \to \infty$ limit of this expression, we can classify a given kernel into one of our three regimes.

The expression we will use gives test MSE in terms of the eigenspectrum of the kernel (as given by the Mercer decomposition) and the eigendecomposition of the target function. The nonnegative eigenvalues $\lambda_1 \geq \lambda_2 \geq ... \geq 0$ and orthonormal eigenfunctions $\{\phi_i\}_{i=1}^{\infty}$ are given by

$$\mathbb{E}_{x' \sim p}[K(x, x')\phi_i(x')] = \lambda_i \phi_i(x), \quad \text{where} \quad \mathbb{E}_{x \sim p}[\phi_i(x)\phi_j(x)] = \delta_{ij}. \tag{2}$$

We note that a kernel must be positive semidefinite and that $\sum_i \lambda_i = \text{Tr}[K] = \mathbb{E}_{x \sim p}[K(x, x)]$, which we assume is finite. Because the eigenfunctions form a complete basis, we are free to decompose the target function as $f^*(x) = \sum_i v_i \phi_i(x)$, where $\{v_i\}_{i=1}^{\infty}$ are eigencoefficients.

The above-mentioned works derive equivalent closed-form expressions for the test MSE of KR in terms of this spectral information using methods from the statistical physics literature. These methods are nonrigorous and rely on approximations (see Appendix A for a discussion), but they are expected to become exact in the large-$n$ limit, and comparison with empirical KR generally confirms a close match even at modest $n$. Here we use the framework of Simon et al. [2021], which expresses the final result in terms of "modewise learnabilities" $\{\mathcal{L}_i\}_{i=1}^{\infty}$, a set of scores in $[0, 1]$ which indicate how well each eigenmode is learned at a given $n$. This choice of variables will simplify our proofs.

Simon et al. [2021] find that test MSE $\mathcal{R}_n$ is approximated by

$$\mathcal{R}_n \approx \mathcal{E}_n \equiv \mathcal{E}_0 \left( \sum_i (1 - \mathcal{L}_i)^2 v_i^2 + \sigma^2 \right), \quad \text{where} \quad \mathcal{E}_0 \equiv \frac{n}{n - \sum_j \mathcal{L}_j^2},$$

$$\mathcal{L}_i \equiv \frac{\lambda_i}{\lambda_i + \kappa}, \quad \text{and} \quad \kappa \geq 0 \text{ satisfies} \quad \sum_i \frac{\lambda_i}{\lambda_i + \kappa} + \frac{\delta}{\kappa} = n. \tag{3}$$

This MSE includes noise on test labels; to instead compute a value for *excess* risk, one would simply subtract $\sigma^2$.

Here we study the asymptotic behavior of $\mathcal{E}_n$ as $n \to \infty$ for varying eigenspectra and ridge values. The fitting regime of the kernel is then given by this limit: (a) if $\lim_{n \to \infty} \mathcal{E}_n = \sigma^2$ (the Bayes-optimal MSE), then fitting is *benign*, (b) if $\lim_{n \to \infty} \mathcal{E}_n \in (\sigma^2, \infty)$, then fitting is *tempered*, and (c) if $\lim_{n \to \infty} \mathcal{E}_n = \infty$, then fitting is *catastrophic*. We obtain conditions on the kernel eigenspectrum under which KR falls into each of these three regimes.

Our proofs rely on several (quite weak) technical assumptions on $\{\lambda_i\}_i$ and $\{v_i\}_i$, the most important of which is that the target function does not place weight in zero-eigenvalue modes (i.e. outside the kernel's RKHS). We defer enumeration and discussion of these conditions to Appendix A, where they are listed as Assumption A.1. Our result is the following:

**Theorem 3.1** (KR trichotomy). *For $\{\lambda_i\}_{i=1}^{\infty}$ and $\{v_i\}_{i=1}^{\infty}$ satisfying Assumption A.1, $\sigma^2 > 0$, and $\mathcal{E}_n$ given by Eq. 3,*

(a) *If $\delta > 0$ or $\lambda_i = i^{-1} \log^{-\alpha} i$ for some $\alpha > 1$, then $\lim_{n \to \infty} \mathcal{E}_n = \sigma^2$.*

(b) *If $\delta = 0$ and $\lambda_i = i^{-\alpha}$ for some $\alpha > 1$, then $\lim_{n \to \infty} \mathcal{E}_n = \alpha \sigma^2$.*

(c) *If $\delta = 0$ and $\lambda_i = i^{-\log i}$, or more generally if $\frac{\lambda_i}{\lambda_{i+1}} \geq \frac{i^{-\log i}}{(i+1)^{-\log(i+1)}}$ for all $i$, then $\lim_{n \to \infty} \mathcal{E}_n = \infty$.*

We defer the proof to Appendix A. The proof proceeds by first showing that asymptotic MSE is dominated by the noise, not the true function, and then computing $\lim_{n \to \infty} \mathcal{E}_0$ in each of the three cases.

Theorem 3.1 can be summarized as follows: a ridge parameter or extremely slow eigendecay leads to benign fitting, powerlaw decay of eigenvalues leads to tempered overfitting, and eigenvalue decay at least as fast as $i^{-\log i}$ leads to catastrophic overfitting. This theorem strongly suggests the satisfying

heuristic that decay *slower* than any $\alpha > 1$ powerlaw is benign, powerlaw decay itself is tempered, and decay *faster* than any powerlaw is catastrophic[8]. As $\alpha$ grows in $(1, \infty)$, powerlaw spectra fully interpolate between benign and catastrophic, suggesting we have not missed any regime of interest to our taxonomy. The fact that the asymptotic MSE from a powerlaw spectrum is simply $\alpha\sigma^2$ is a pleasant surprise.

Theorem 3.1 has several consequences for KR with familiar kernels and the training of infinite-width networks. For illustrative purposes, we contrast the Gaussian (RBF) kernel $K_G(x_1, x_2) = e^{-w^{-2}\|x_1-x_2\|_2^2}$ with the Laplace kernel $K_L(x_1, x_2) = e^{-w^{-1}\|x_1-x_2\|_2}$, where $w$ is a bandwidth parameter. With data drawn from a $d$-dimensional manifold, the Gaussian and Laplace kernels have eigenspectra that decay like $\lambda_i \sim e^{-i^{2/d}}$ (as can be seen by taking a Fourier transform of $K_G$) and $\lambda_i \sim i^{-(d+1)/d}$ [Bietti and Bach, 2020], respectively. We note that ReLU NTKs, restricted to the hypersphere, have the same eigendecay as the Laplace kernel [Chen and Xu, 2020, Geifman et al., 2020]. The implications of Theorem 3.1 include the following:

- KR with a fixed positive ridge parameter will fit any function in the kernel's RKHS benignly as $n \to \infty$.

- Ridgeless KR with *Laplace kernels* or *ReLU NTKs* will exhibit *tempered overfitting* with $\Theta(1/d)$ excess MSE, approaching benignness as dimension grows.

- Ridgeless KR with the *Gaussian kernel* will exhibit *catastrophic overfitting*.

- The asymptotic behavior of KR with non-ReLU NTKs *depends on the activation function*. Virtually any kernel on the $d$-sphere (including the Gaussian kernel) can be realized as the NTK of a wide network with a proper choice of activation function [Simon et al., 2022], and thus there exist choices of activation function that yield catastrophic as well as tempered overfitting.

- Early stopping is known to act as an effective ridge parameter for wide networks [Ali et al., 2019], and thus we should expect that early-stopped wide networks will fit benignly.

We provide experiments illustrating Theorem 3.1 with Gaussian and Laplace kernels in Section 4. We additionally check Theorem 3.1b with several subsequent experiments: Figure 5 shows that, in synthetic KR with Gaussian eigenfunctions and exact powerlaw spectra, $\lim_{n\to\infty} \mathcal{E}_n$ is $\alpha\sigma^2$. And Figure 6 in Appendix B shows that, as predicted, Laplace kernels indeed appear to have asymptotic risk that decays like $\Theta(1/d)$.

We note that, to our knowledge, no well-known kernel has an eigendecay slower than all $\alpha > 1$ powerlaws in finite dimension, and finding one in closed form — yielding ridgeless KR which overfits benignly — is an interesting open problem.

## 4  Experiments

Having demonstrated the three types of fitting theoretically in KR, we now present a series of experimental results illustrating these regimes in both KR and deep neural networks (DNNs). We provide full experimental details in Appendix C.

### 4.1  Experiments on Kernel Regression

In Figure 4, we run KR on the following synthetic data distribution: the inputs $x$ are sampled from the unit sphere $\mathcal{S}^{d-1}$, and the targets $y$ are zero-mean Gaussian noise ($y \sim \mathcal{N}(0, 1)$). This is an extremely simple regression setting: we are just trying to learn the constant-0 function under Gaussian observation noise. We run KR with three choices of kernel: (A) Gaussian kernel with ridge, (B) Laplace kernel without ridge, and (C) Gaussian kernel without ridge. Figure 4 shows that as we increase the sample size, these three settings exhibit benign, tempered, and catastrophic behavior. This agrees with the spectral predictions of Theorem 3.1[9].

---

[8]We leave a complete proof of this heuristic as an open problem.

[9]Though not reported here, we find the ridged Laplace kernel also exhibits benign fitting as expected.

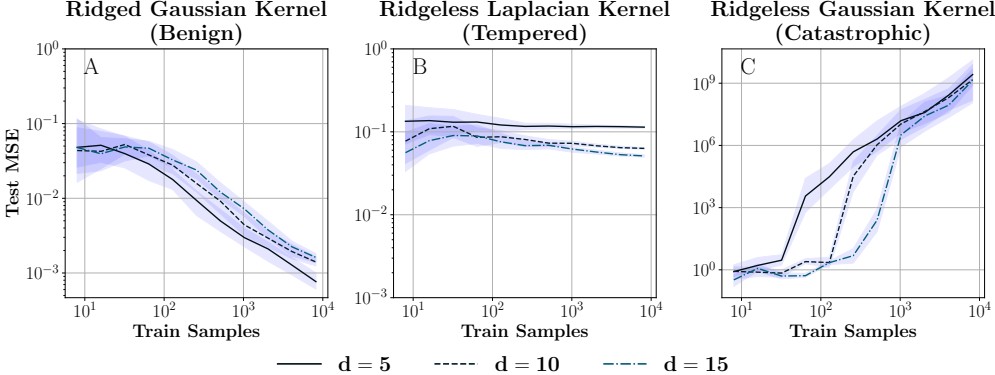

Figure 4: **Kernel regression can exhibit all three fitting regimes with proper choice of ridge parameter and kernel.** Plots show learning curves for KR with data $\{x_i\}$ sampled uniformly from the unit sphere $\mathcal{S}^{d-1}$, trained with pure noise target labels $y_i \sim \mathcal{N}(0,1)$. Test MSE is computed with respect to a clean test set. **(a)** KR with a Gaussian kernel and nonzero ridge is asymptotically benign. A ridge value of $\delta = 0.1$ was used. **(b)** Ridgeless KR with a Laplace kernel exhibits tempered overfitting. **(c)** Ridgeless KR with a Gaussian kernel exhibits catastrophic overfitting.

In Appendix B, we report KR experiments using synthetic kernels with exact powerlaw spectra trained on noisy data. We find that as the spectral decay $\alpha$ varies, the (asymptotic) test MSE is indeed approximately $\alpha\sigma^2$ in agreement with part (b) of Theorem 3.1.

## 4.2 Experiments on Deep Neural Networks

**Interpolating DNNs.** Figure 2 and Figure 8a show the noise profiles for ResNets trained to interpolation on *two-class* and *ten-class* versions of CIFAR-10, respectively. In both settings, interpolating DNNs do not approach Bayes optimality, and instead exhibit tempered overfitting. This tempered behavior is widespread across even much simpler DNN settings. For example, in Figure 8b, we train a three-layer interpolating MLP for binary classification on an extremely simple synthetic dataset: with inputs drawn from $\mathcal{S}^9$ and ground-truth labels as the constant function $f^*(x) = 1$. Even in this simple setting, at large sample size, interpolating DNNs are not benign—they do not successfully learn the constant function in the presence of even slight label noise. Note that although this experiment is outside the technical scope of Theorem 3.1, it is heuristically consistent: wide ReLU MLPs tend to have NTKs with powerlaw spectra, and thus will exhibit tempered overfitting when trained in the NTK regime.

**Early-stopped DNNs.** We now consider DNNs that have been optimally early-stopped. Figure 9 shows noise profiles for Wide ResNets trained on a binary version of SVHN, both *stopped early* (Figure 9a) and *trained to interpolation* (Figure 9b). The early-stopped ResNets approach benign fitting as $n$ grows, with low error even at sizable noise levels, while the ResNets trained to interpolation quickly converge to a tempered noise profile. This mirrors the behavior of MLPs on binary MNIST, after one epoch of training and after interpolation, shown earlier in Figure 3. Although these benign DNN results are outside the formal scope of known theoretical results, it is heuristically consistent with results such as Ji et al. [2021], which show that certain wide and shallow ReLU MLPs are consistent when early-stopped.

The above discussion suggests that as a single DNN is trained, it exhibits benign fitting early in training (when it has not fit the noise in its train set), and then transitions to tempered overfitting late in training (as it eventually fits the noise). In Appendix E, we show a simple experiment in which an MLP trained on synthetic noisy data clearly exhibits this transition between regimes.

## 5 Limitations

In this paper, we have presented a taxonomy of overfitting, presented empirical evidence that DNNs trained to interpolation exhibit tempered overfitting, and identified spectral conditions under which

KR, a toy model for DNNs, falls into each regime. This is a first study into this regime, and many questions await careful exploration. We detail several here.

First, our DNN results are entirely empirical, and there is room for complementary theoretical studies into the overfitting regimes of shallow networks and wide networks with NTK and mean-field parameterization. Second, while we have used several real and synthetic datasets, all are of at most moderate size, and a study at large scale — using correspondingly large models — is potentially interesting. Third, we have found that in some realistic settings, interpolating DNNs are tempered, but it remains open whether there might exist settings or tasks for which interpolating DNNs overfit benignly or catastrophically. Fourth, Theorem 3.1 required a "universality" assumption which ought to be checked for each kernel. Finally, our KR results suggest that input or manifold dimension should play a role in the degree of tempered overfitting, and the effect of dimensionality on overfitting is yet to be disentangled.

## 6   Conclusion

In this paper we study the nature of overfitting in learning methods which interpolate their training data. Much of contemporary theory and experimental work categorizes overfitting as either catastrophic or benign. In contrast, we observe that many natural learning procedures, including DNNs used in practice, overfit in a manner that is neither benign nor catastrophic— but rather in an intermediate regime. We identify and formally define this regime, which we call *tempered overfitting*. We present empirical evidence of learning procedures that exhibit tempered overfitting, on both synthetic and natural data, using kernel machines and deep neural networks. We show tempered overfitting can be quantified in terms of noise profiles, which measure how asymptotic performance depends on noise in the train distribution. For kernel regression, we provide a theoretical result in the form of a trichotomy: conditions on problem parameters which yield each of the three regimes of overfitting.

Our work presents an initial study of tempered overfitting and lays the framework for future study of what we believe is a rich and relevant regime for modern learning. We hope this framework inspires further investigation into tempered overfitting for more complex models, both theoretically and experimentally. For example, it is open to understand which conditions on neural network architecture, training hyperparameters and data distribution lead to benign, tempered, or catastrophic overfitting, in analogy to our "kernel trichotomy", and an answer might shed light on practical DNNs.

**Acknowledgments**

The authors thank Nikhil Ghosh, Annabelle Carrell, Spencer Frei, Daniel Beaglehole, Gil Kur, and Russ Webb for useful discussion and feedback on the manuscript. JS additionally thanks two strangers on a DC Metro for useful discussion. PN especially thanks the Simons Institute, UCSD, and Apple MLR for collaborative environments which made this project possible.

We are grateful for support from the National Science Foundation (NSF) and the Simons Foundation for the Collaboration on the Theoretical Foundations of Deep Learning[10] through awards DMS-2031883 and #814639 as well as NSF IIS-1815697 and the TILOS institute (NSF CCF-2112665). NM is thankful to be funded and supported for this research by the Eric and Wendy Schmidt Center. JS gratefully acknowledges support from the NSF Graduate Fellow Research Program (NSF-GRFP) under grant DGE 1752814. This work used the Extreme Science and Engineering Discovery Environment (XSEDE) [Towns et al., 2014], which is supported by NSF grant number ACI-1548562, Expanse CPU/GPU compute nodes, and allocations TG-CIS210104 and TG-CIS220009.

**Author Contributions**

NM developed and performed the bulk of the experiments, drafted the initial version of the manuscript, actively contributed to writing and editing, and led the team logistically. JS proposed and proved the theoretical KR results, aided in experimentation, and contributed to the writing throughout the paper. AA developed and executed most of the kernel experiments, provided support to other experiments, and helped edit the manuscript. PP and MB provided guidance for experiments and theory, and reviewed and edited drafts of the manuscript. PN initiated this research agenda and provided active input throughout the experimentation, theory, and writing phases of the project.

---

[10]https://deepfoundations.ai/

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
