# OpenReview forum: "Benign, Tempered, or Catastrophic: Toward a Refined Taxonomy of Overfitting"
_NeurIPS.cc/2022/Conference — NeurIPS 2022 Accept_

### Official Review · Reviewer_HmMk · 2022-06-14

**Rating:** 4
**Confidence:** 3
**Soundness:** 2 fair
**Presentation:** 2 fair
**Contribution:** 3 good

**Summary:**

Summary.

The paper is dedicated to investigating the overfitting phenomenon of overparameterized neural networks. The authors argue that real interpolating methods like deep networks do not fit benignly, although benign overfitting has been widely instructive to study. They name the behavior as tempered overfitting and demonstrate it via kernel regression and deep neural network experiments.

**Questions:**

Refer to the weakness section.

**Limitations:**

The checklist indicates that the authors do not discuss the limitation and societal impact.

**Strengths And Weaknesses:**

Pros.

1. This paper is well-organized.

2. The formulated problem of tempered overfitting is interesting.

3. Both theoretical and empirical analyses are conducted.

Cons.

1. Poor language "We first kernel (ridge) regression, obtaining conditions on the ridge parameter and kernel eigenspectrum under which KR exhibits each of the three behaviors, thus obtaining a relatively complete picture of overfitting for kernel regression."

2. The term "catastrophically overfitting" is ambiguous. It seems the authors mean the standard overfitting in deep learning. Since continual learning (https://www.cs.uic.edu/~liub/lifelong-learning/continual-learning.pdf) and fast adversarial training (https://arxiv.org/abs/2105.02942), there are other definitions of the "catastrophically overfitting".

3. The tempered overfitting is a very general concept but only validated in limited cases. As we know, different networks have distinctive optimization behaviors. The paper's experiments are carried out only on ResNets on CIFAR and SVHN. More network backbones like VGG, MobleNet, and more datasets like ImageNet are needed.

4. What are the corresponding descriptions or phenomena of piecewise linear interpolation in the multi-dimension cases?

---

> ### Author Response · Authors · 2022-08-02
> **Thank you for your feedback**
>
> We thank the reviewer for their feedback. Here we address their concerns one by one:
>
> 1. Thank you for noting this typo. We have thoroughly vetted our manuscript for typos and smoothed over our presentation. Please see our updated version.
> 2. On catastrophic forgetting in continual learning: this is a distinct behavior that does not relate to overfitting.
> 3. While it is true we only run limited experiments, this is appropriate for a first study into a new regime. Given that our paper identifies a new concept and contains novel theoretical results validating it, the fact that our experiments “only” use DNNs + ResNets on synthetic datasets + standard image datasets is quite reasonable. That said, we agree further experimental validation is an important direction for this line of work, and as a first step, we have added a new experiment training VGG networks on CINIC-10 to the present manuscript (Appendix C.3, Figure 8). CINIC-10 is a combination of CIFAR-10 and Tiny ImageNet examples, and thus is more difficult than CIFAR-10 but easier than ImageNet, which makes it computationally reasonable still to interpolate at high label noise.
> 4. While piecewise linear interpolation does not generalize unambiguously to higher dimensions, one similar method in 2D is bilinear interpolation (or multilinear interpolation in higher dimension). Like piecewise linear interpolation, this method exhibits tempered overfitting, though this fact is complicated by its requirement that training points lie on a mesh. Though interesting, discussion of these methods lies outside the scope of the present work. Simplicial interpolation - which appears in our Table 1 as a tempered method - is another such generalization of piecewise linear interpolation.
>
> Finally, we must note our feeling that the reviewer’s listed concerns - even if they all applied as stated - are not big enough to warrant the low score given, and can all easily be addressed with minor rewriting or additional experiments. If the reviewer feels they have no outstanding vital concerns regarding our paper, we respectfully ask that they increase their score accordingly.

---

### Official Review · Reviewer_PuoQ · 2022-07-04

**Rating:** 6
**Confidence:** 3
**Soundness:** 3 good
**Presentation:** 4 excellent
**Contribution:** 3 good

**Summary:**

This paper proposes a taxonomy of overfitting based on the generalization error when the sample size goes to infinity.
Specifically, the authors study the case of kernel regression and derive the asymptotic generalization gap in some cases.
Based on these cases, the authors show that (a) kernel regression with regularization is benign; (b) Some commonly-used kernels such as ReLU NTK and Laplace KR are tempered; and (c) Gaussian KR is Catastrophic.

Overall, this paper provides some interesting new ideas for the deep learning community and therefore has the potential to be accepted.


**Questions:**

See the above Flaws part.

**Limitations:**

No potential negative societal impact.

**Strengths And Weaknesses:**

Contributions.
1. The authors provide a new taxonomy of overfitting based on how the asymptotic generalization error performs.
2. The authors derive generalization errors for some cases of kernel regression and show which category these cases belong to.
3. The authors show that ReLU-based NTK falls in the Tempered case.

Flaws:
It is still unclear how this taxonomy helps in deep learning. Could the authors provide more information on what if we know this taxonomy in practice?

Typos:
Line2: be able *to*
Line13: we first kernel regression?

---

> ### Author Response · Authors · 2022-08-02
> **Thank you for your feedback**
>
> Thank you for the review. Regarding your main concern on the importance of our taxonomy, the community of interest for this paper is primarily deep learning theorists, for whom our work clarifies an important and ongoing conversation regarding the nature of overfitting in deep networks. Benign overfitting is a hot topic and the leading conceptual framework for understanding why, despite intuitions from statistical learning theory, overparameterized neural networks can generalize at all. To see its importance, note that two papers on this topic - “Benign Overfitting in Linear Regression” (Bartlett et al. ‘19) and “Surprises in High-Dimensional Ridgeless Least Squares Interpolation” (Hastie et al. ‘19) - each have ~400 citations. Our paper is another step in this line of work. These works, like ours, are aimed at understanding deep learning, not directly aiding practice. The hope of the field is of course that understanding will, in the long run, greatly help deep learning in practice.
>
> That said, our work does indeed have lessons for practitioners, including the following:
> * The fact that DNNs are often *tempered* implies that noise in the training set will reliably cause noise in the test set, and so clean data is very important, and noisy data is not easily fixed by having more data.
> * The noise profile plot format we use is potentially useful for practitioners interested in the effect of various types of noise on DNN performance.
> * Our kernel results suggest that a DNN’s robustness to noise is strongly affected by its activation function.
>
> We have fixed spelling and grammar errors as reported. If you have outstanding concerns, we would be happy to hear and address them. If we have addressed your concerns, we ask that you please consider raising your score accordingly.

---

> > ### Comment · Reviewer_PuoQ · 2022-08-09
> > **Thank you for your response**
> >
> > Thank you for your response. And I would like to keep my score unchanged.

---

### Official Review · Reviewer_xyxV · 2022-07-10

**Rating:** 8
**Confidence:** 3
**Soundness:** 4 excellent
**Presentation:** 4 excellent
**Contribution:** 4 excellent

**Summary:**

As the title already suggests, the paper proposes an extended taxonomy of overfitting, adding an additional category of “tempered” overfitting, to the previously acknowledge benign and catastrophic forms.  Motivated by the increased study of overparametrized neural networks, various existing learning procedures are first categorized into the respective three categories from a regression and classification perspective. After application of the taxonomy, two specific cases are studied further, first kernel regression, in theory and practice, and then deep neural networks form an empirical standpoint.

**Questions:**

If I understand correctly, the checklist is suggesting that the authors do not wish to publish their code (please correct me if I am wrong here). I wonder what the rationale behind this decision is and whether the authors would perhaps reconsider to also open-source in favor of reproducibility? Given that there are some assumptions at play, perhaps some that we discover only later, it would be intriguing to facilitate replication of the experiments.

**Limitations:**

Please see first weakness point, which targets limitations and already includes constructive suggestions for improvements.

**Strengths And Weaknesses:**

**Strengths**

* The paper is exceptionally well written. Whereas there are many nicely written and structured papers, the presented manuscript clearly surpasses the average presentation. The introduction starts with a crystal clear motivation, an intuitive example, and the contributions are outlined appropriately. The related work section manages to not only give credit to a myriad of prior efforts,  but also to pinpoint the key differences and how the work builds on top. Although I am personally not familiar with various of the cited works and some of the presented topics, I never started to feel lost. In parts, this is further due to the fact that the writings strikes a balance between presenting theoretical statements and also following up with intuition and more verbose statements. Similarly, I appreciated that the text clearly emphasize when a statement has been shown/proven, versus e.g. when a conjecture is based on what the authors refer to as “experimental folklore”.
*  Subjectively, I tend to dislike footnotes, based on the rationale that footnotes are either important enough to mention directly in the text, or omit them entirely. However, in this work, I actually found them quite helpful and appreciated the additional small notes at the bottom.
* The taxonomy itself, examples of table 1, and the follow-up theoretical and empirical work have the prospect of facilitating future discussions about overfitting, overparametrization and the various ways in which ML algorithms learn. In some sense, the paper is almost by definition limited in scope from an experimental angle, given that the topic is relevant and is concerned with the study of a plethora of models & learning procedures. With this in mind, the empirical results are quite convincing, not only on the sense that the evidence experimentally supports the presented statements/theory, but also because it spans multiple datasets, as well as simple & more complicated methods.

**Weaknesses**

* My only key weakness, on the one hand because of my somewhat limited expertise in the theoretical aspects of the paper, on the other hand precisely because of it, is the lack of a discussion on limitations. Particularly, I wish there existed a concise separate limitation section, such that the perhaps more generally informed ML reader can directly get an overview of what to take for granted and where to still be particularly cautious. To emphasize, I believe the paper is to an extent already adequately presenting this information, e.g. by pointing out that the DNN parts are yet to be theoretically further explored/supported similarly to the simpler perspective of KRR. Having a dedicated place to consult would however be additionally helpful.
* This point is perhaps less of a weakness and more of a suggestion for further improvement: After reading the text and looking at the plots in figure 5, it took me a bit to grasp whether the statement on tempered vs. benign overfitting in vs. early-stopped DNNs  is as clearly exhibited as suggested by the text. Intuitively it does become apparent as the right-hand side plot does not illustrate the dependency on “n”, as the left one does. However, in absolute terms it’s also somewhat tough to compare the two plots side by side. Perhaps a single larger plot with two different colors could more clearly show the discrepancy.

Overall I believe there are many strengths to the paper and it will be very beneficial to disseminate the work to the NeurIPS community.

**Minor**

* Figure 3 is missing a t in “catastrophic” in the title.
* The references are a bit inconsistent, some including URLs & some not, sometimes including conference dates or venues, in parts referring to proceedings vs. just the conference etc. An extra pass to make them more consistent, in whichever preferred way, would be nice.

---

> ### Author Response · Authors · 2022-08-02
> **Thank you for your feedback**
>
> Thank you for the careful review of our work. We’re gratified you found our message important and well-supported and our literature survey clear. We have uploaded a revised manuscript containing even more experimental support using CINIC-10 and VGG networks (Appendix C.3, Figure 8).
>
> We agree with your minor critiques; we have now carefully checked our manuscript for typos, and we will perform a pass over the references for the final version.
>
> Regarding the limitations of our work, we like this idea, and we have added a Limitations section in Appendix D detailing what can be confidently concluded from our study and what requires more experimental support. We plan to shift this section into the main-text for the camera ready so that it is more prominent. Here is the text of this paragraph:
>
> In this paper, we have presented a taxonomy of overfitting, presented empirical evidence that DNNs trained to interpolation exhibit tempered overfitting, and identified spectral conditions under which KR, a toy model for DNNs, falls into each regime. This is a first study of this regime, and many questions await careful exploration. We detail several here.
>
> First, our DNN results are entirely empirical, and there is room for complementary theoretical studies into the overfitting regimes of shallow networks and wide networks with NTK and mean-field parameterization. Second, while we have used several real and synthetic datasets, all are of at most moderate size, and a study at large scale --- using correspondingly large models --- is potentially interesting. Third, we have found that in some realistic settings, interpolating DNNs are tempered, but it remains open whether there might exist settings or tasks for which interpolating DNNs overfit benignly or catastrophically. Fourth, Theorem 1 required a “universality” assumption which ought to be checked for each kernel. Finally, our KR results suggest that input or manifold dimension should play a role in the degree of tempered overfitting, and the effect of dimensionality on overfitting is yet to be disentangled.
>
>
> On Figure 5, we have rerun this experiment with different parameters, yielding a more convincing plot, and also rewritten the caption, adding an explanation of each subplot:
>
> > “(a) With early stopping, noise profiles approach zero test error as $n$ increases, even with finite label noise, indicating benign fitting.
> > (b) When training to interpolation, noise profiles converge to a limiting, roughly linear curve as $n$ increases, indicating tempered fitting.”
>
> Regarding our code, we fully intend to publish the code for our DNN experiments and have adjusted the checklist accordingly. We have already submitted our code in the supplements.
>
> Thank you again for the favorable review.

---

> > ### Comment · Reviewer_xyxV · 2022-08-04
> > **Thank you for the response and clarifications**
> >
> > I appreciate the response and the fact that the authors have taken my suggestions into account in their various manuscript revisions.
> >
> > As stated in my initial review, I believe this work to be very interesting and helpful to the community.
> >
> > Given the rebuttal, I intend to keep my high score for this submission.

---

### Meta-Review · Area_Chair_N5ii · 2022-08-26

**Recommendation:** Accept
**Confidence:** Less certain

**Metareview:**

This paper proposes a new taxonomy of overfitting that has the prospect of facilitating future discussions about overfitting, overparametrization and various mysteries of deep learning. The meta-reviewer recommends acceptance.

**Award:**

No

---

### Decision · Program_Chairs · 2022-09-14

Accept